# Effects of stimulus onset asynchrony on cognitive control in healthy adults

**Abdolvahed Narmashiri** [1,2]*

1 Institute for Research in Fundamental Sciences (IPM), School of Cognitive Sciences, Tehran, Iran,
2 Electrical Engineering Department, Bio-Intelligence Research Unit, Sharif Brain Center, Sharif University of Technology, Tehran, Iran

* A_narmashiri@ipm.ir

## Abstract

The efficiency of cognitive control in healthy adults can be influenced by various factors, including the stimulus onset asynchrony (SOA) effect and strategy training. To address these issues, our study aims to investigate the impact of SOA on single-mode cognitive control using the Go/No-Go task, as well as the manifestation of proactive control within dual mechanisms of cognitive control through the AX-CPT task. In single-mode cognitive control, extending SOA led to significantly enhanced reaction times (RTs) during Go trials, suggesting improved task performance with increased preparation time. Moreover, the analysis revealed consistently higher accuracy rates in No-Go trials than to Go trials across all SOA levels, indicating robust inhibition processes unaffected by SOA variations. In the dual mechanisms of cognitive control, significant variations in RT and accuracy were observed among different trial types. Notably, participants exhibited superior performance in detecting targets during BY trials and shorter RTs in BX trials, indicative of efficient processing of target stimuli. Conversely, prolonged RTs in AY trials suggest proactive control strategies aimed at maintaining task-relevant information and inhibiting irrelevant responses. Overall, these findings highlight the effect of SOA on single-mode cognitive control and the emergence of proactive control within dual mechanisms of cognitive control in healthy adults.

## Introduction

Cognitive control, comprising both single-mode and dual mechanisms, serves as a fundamental aspect of executive functioning, allowing individuals to regulate behavior and decision-making processes [1–3]. Single-mode cognitive control refers to a uniform strategy used to manage cognitive processes, often assessed through tasks like the Go/No-Go paradigm, which involves the ability to inhibit automatic responses and maintain task goals amidst distracting stimuli [4–7]. Conversely, dual mechanisms of cognitive control encompass two distinct modes: proactive control, which involves the anticipatory allocation of cognitive resources, and reactive control, which involves responding to immediate demands. These dual mechanisms are examined in tasks like the AX-Continuous Performance Task (AX-CPT), which requires the interplay of proactive and reactive strategies to optimize performance in

**Competing interests:** The author have declared that no competing interests exist.

challenging cognitive contexts [3,8]. The efficiency of cognitive control in healthy adults can be influenced by various factors, including the stimulus onset asynchrony (SOA) effect and strategy training.

The SOA refers to the temporal interval between the presentation of two stimuli in a cognitive task [9]. Research suggests that SOA can significantly impact cognitive functions, including inhibition [10]. For instance, shorter SOAs may enhance response inhibition by reducing the time available for automatic response activation, whereas longer SOAs may provide individuals with more time to prepare and execute inhibitory responses [11]. Furthermore, SOA variations can affect the allocation of attentional resources, which in turn can modulate inhibition processes [12]. Studies have investigated how different SOA durations influence inhibitory control, shedding light on the underlying mechanisms and neural correlates involved [13]. Moreover, neuroimaging techniques such as functional magnetic resonance imaging (fMRI) have been used to examine the neural basis associated with SOA effects on inhibition, offering insights into the intricate interplay between temporal dynamics and cognitive processes [14].

In addition to the role of SOA in single-mode cognitive control, recent studies have underscored the significance of strategy training, particularly in light of the emergence of proactive control within the dual mechanisms of cognitive control in healthy adults [15]. Proactive control refers to the anticipation and implementation of cognitive strategies to prepare for upcoming task demands, enabling individuals to regulate their behavior proactively and optimize performance [2,16]. This proactive engagement complements reactive control mechanisms, allowing individuals to flexibly adapt to varying cognitive demands and environmental contexts [15]. Studies indicate that proactive control is important in the cognitive control dual mechanisms among healthy adults. Previous research has investigated the neural substrates and behavioral manifestations of proactive control, providing insights into its role in cognitive flexibility and adaptive behavior [15,17,18]. However, further studies are needed to confirm the specific training strategy within dual-mechanism cognitive control in healthy adults.

Despite the substantial body of research in this area, a significant gap persists in our understanding of the precise impact of Stimulus Onset Asynchrony (SOA) on cognitive control within the Go/No-Go task. While some studies have indicated clear modulatory effects of SOA on inhibitory performance [19], there remains a need for systematic investigations to elucidate the intricate relationship between SOA and inhibition in Go/No-Go paradigms. These efforts are essential for advancing our understanding of cognitive control mechanisms [20,21] and for identifying optimal conditions that facilitate effective decision-making and adaptive responses. Furthermore, our study's confirmation of previous findings regarding the role of proactive control within the dual mechanisms of cognitive control during AX-CPT paradigms in healthy adults underscores its significance in elucidating cognitive flexibility and adaptive behavior across various cognitive tasks and contexts. Therefore, a rigorous and methodical approach to exploring these dimensions not only addresses current gaps in knowledge but also provides a structured framework for advancing our understanding of cognitive control dynamics and their practical implications.

## Materials and methods

### Participants

Thirty-four healthy students (19 females and 15 males), who were determined to be right-handed using the Edinburgh Handedness Inventory, were recruited for this research. Participants were selected based on their availability and willingness to participate. Exclusion criteria included self-reported history of psychosis, mental disorders, acute or chronic illnesses,

neurological or personality issues, substance abuse, or epilepsy. Participants also reported normal or corrected-to-normal vision. Power analysis using G*Power 3.1 [22] with a medium effect size, power of 0.85, and α = 0.05, suggested a sample size of thirty-one, with slight oversampling to mitigate potential technical issues. Recruitment took place through advertisements on campus and social media platforms. Participants were compensated with gifts for their involvement. In this study, supported by the Institute for Cognitive Science Studies, no clinical trials were conducted, and no interventions were administered. Instead, participants completed two computerized cognitive tasks after providing written informed consent before enrollment. The study adhered to the principles outlined in the Declaration of Helsinki.

## Stimuli

**Cue-dependent Go/No-Go task.**   This task assesses cognitive control in a specific mode, emphasizing the predictive elements of both inhibitory and activational control systems. Prior cues inform participants about the upcoming type of stimulus (go or stop) with high accuracy. Participants' inhibitory and activational tendencies adjust rapidly based on these cues, initiating preparatory actions for either restraining or executing a response [23,24]. The conditions with go cues are especially intriguing as they prompt a strong readiness to respond, resulting in quicker reactions to go targets. Yet, participants need to counteract this readiness to suppress their response when presented with a no-go target. Interestingly, there are more instances of failure to inhibit responses to no-go targets after go cues compared to after no-go cues, indicating a heightened challenge in inhibiting pre-existing responses [24]. Moreover, the ability to exert inhibitory control in the presence of prepotent go cues seems significantly affected by psychoactive drugs, encompassing both stimulants and depressants [25]. The study employed Inquisit software (version 6.6.1) running on a personal computer. Each trial followed a specific sequence: (a) a fixation point (+) shown for 800 ms; (b) a blank white screen displayed for 500 ms; (c) presentation of a cue with one of five stimulus onset asynchronies (SOAs = 100, 200, 300, 400, and 500 ms); (d) appearance of a go or no-go target, remaining visible until a response occurred or 1000 ms had passed; and (e) a 700 ms intertrial interval.

The cue, which measured 7.5 × 2.5 cm, was shown as a black-outlined rectangle (0.8 mm thick) centrally displayed on a white background of a computer monitor. This cue could appear either horizontally (2.5 × 7.5 cm) or vertically (7.5 × 2.5 cm). Go targets, colored green, and no-go targets, colored blue, were displayed as solid colors within the rectangle cue. Participants were instructed to press the forward slash (/) key on the keyboard upon seeing a green go target and to refrain from responding when a blue no-go target appeared. Participants used their preferred hand's index finger to respond. Go and no-go targets were presented in different colors, and the cue's orientation indicated the probability of encountering either type of target. Vertical cues suggested an 80% chance of encountering a go target and a 20% chance of a no-go target, while horizontal cues indicated an 80% probability of encountering a no-go target and a 20% probability of encountering a go target. Therefore, vertical cues indicated a need for a go response, while horizontal cues signaled a requirement for a no-go response. Changing SOAs between cues and targets encouraged participants to focus on the cues, and the random nature of SOAs prevented anticipation of when the targets would appear. The experiment consisted of 250 trials covering all possible combinations of cues and targets. There were an equal number of vertical and horizontal cues (125 each), each preceding an equal number of go and no-go target stimuli (125 each). Every cue-target combination was presented at each of the five SOAs, with an equal distribution of SOAs between pairs. The order of presentation for cue-target pairs and SOAs was randomized. The computer recorded responses during each trial,

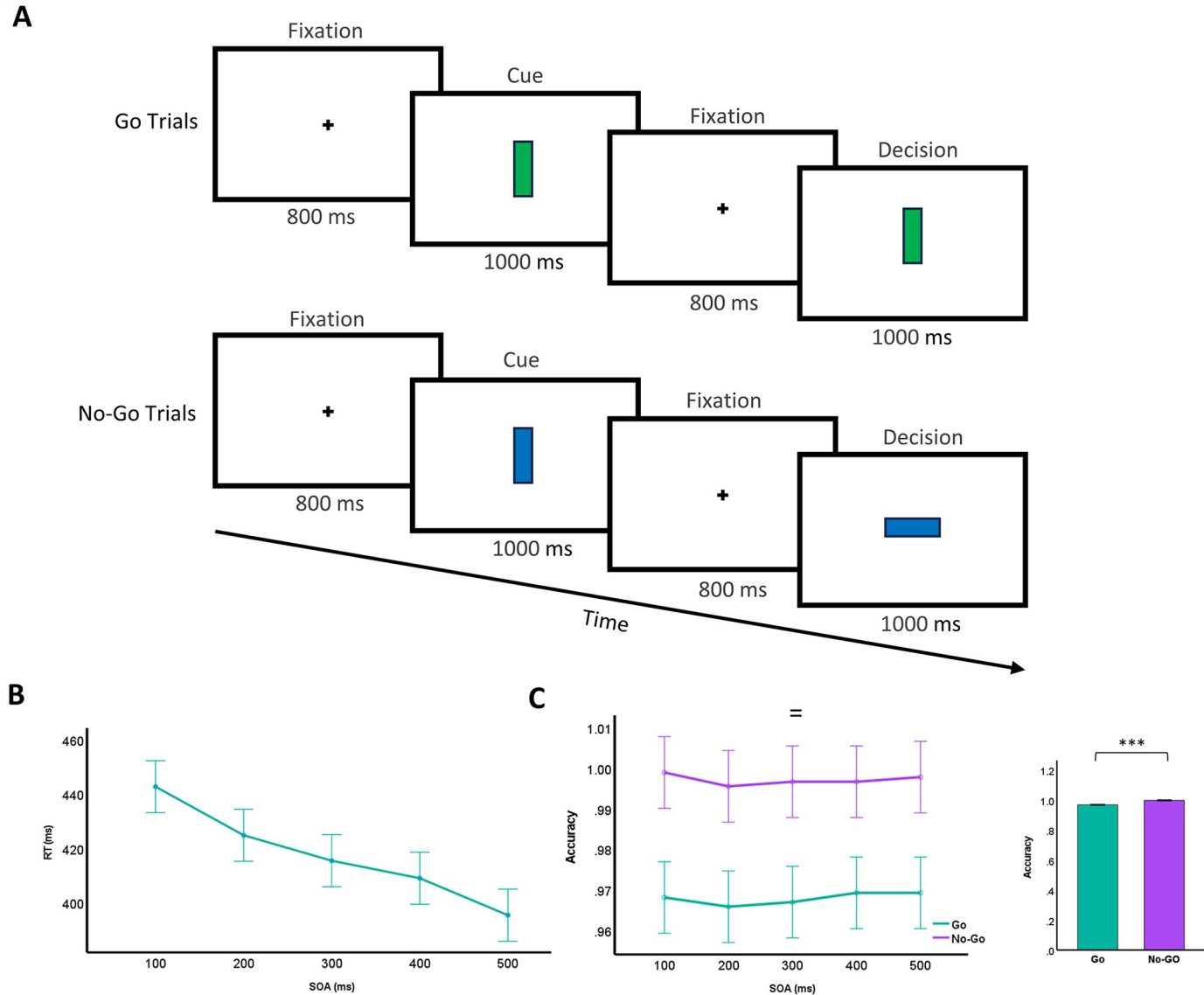

**Fig 1. Single-mode cognitive control paradigm, task design, and behavioral performance. A)** View of Go/No-Go task that show the procedure of Go and (top) No-Go trials (bottom). Each trial followed a specific sequence: A fixation point (+) displayed for 800 ms; a blank white screen lasting 500 ms; presentation of a cue for one of five stimulus onset asynchronies (SOAs = 100, 200, 300, 400, and 500 ms); appearance of a go or no-go target, remaining visible until a response occurred or 1000 ms had elapsed; and an intertrial interval of 700 ms. B) RT (ms) as functions of SOAs (ms) for Go trials (depicted in green). C) Accuracy as functions of SOAs (ms) for Go trials (depicted in green) and No-Go trials (represented in purple). Bar plot shows accuracy in all SOAs for Go trials (green) and No-Go trials (purple). The symbols *, **, and *** indicate statistical significance levels at p < 0.05, p < 0.01, and p < 0.001, respectively. The symbols =, /, and x represent the main effects of search efficiency, set size, and their interaction, respectively.

measuring reaction times (RT) from target onset to key press in ms. On average, completing the test took approximately 10 minutes (Fig 1A).

The task design incorporates various SOA values (100, 200, 300, 400, and 500 ms) strategically chosen to probe different aspects of cognitive processing [23,24]. Shorter SOAs (100 and 200 ms) typically assess automatic or pre-attentive processing, where stimuli are closely spaced, allowing minimal time for deliberate attentional shifts. Intermediate SOAs (300 and 400 ms) explore more controlled processing, balancing between automatic and effortful attentional

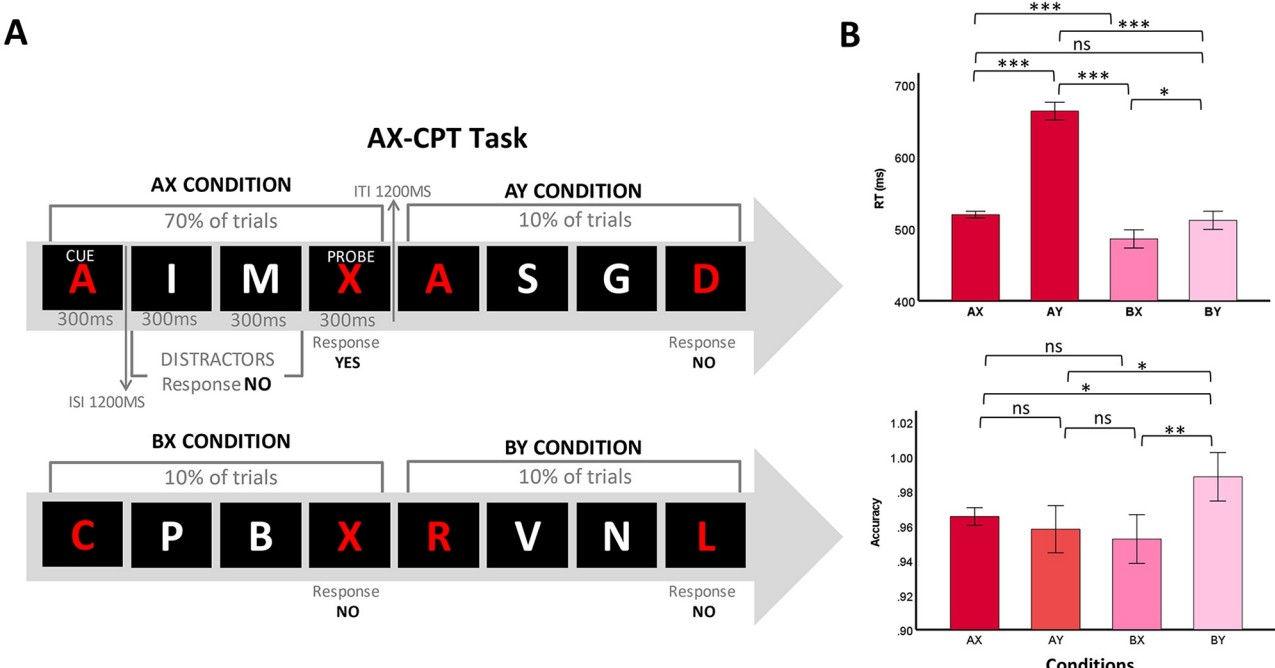

**Fig 2. Dual mechanism of cognitive control paradigm, task design, and behavioral performance.** A) View of AX-CPT task. Participants completed an AX-CPT task, viewing letter trials lasting 300 ms on a black screen, with a 4200 ms interval between cue and probe presentation. They were instructed to remember the cue ("A" or any letter except "X," "K," and "Y") until they saw the probe ("X" or any letter except "A," "K," or "Y"). If the cue was "A" followed by the probe "X," they pressed the "yes" button; otherwise, they pressed the "no" button. B) RT (top) and accuracy (bottom) as functions of conditions (X, AY, BX, and BY trials). The symbols *, **, and *** indicate statistical significance levels at p < 0.05, p < 0.01, and p < 0.001, respectively.

mechanisms. Longer SOAs (500 ms) emphasize sustained attention and decision-making processes, reflecting scenarios where stimuli are presented further apart, requiring sustained attention to maintain task performance. This range of SOAs enables the examination of how temporal factors influence cognitive processing speed, accuracy, and the efficiency of executive functions such as attentional control and response inhibition.

**AX-continuous performance task (AX-CPT).** AX-CPT is utilized as a behavioral measure to assess the dual mechanisms of control (DMC, Fig 2A) [26,27]. For this procedure, participants were presented with a sequence of letters for 300 ms each in the center of a black screen. The letters were displayed on a cue-probe basis so that 4,200 ms elapsed between presentation of cue and probe. The intertrial interval was 1,200 ms. Participants were instructed to maintain the cue in memory (either the letter "A" or any other letter except "X," "K," and "Y," due to their perceptual similarity with "X") until they saw the probe (either the letter "X" or any other letter except "A," "K," or "Y"). Whenever they saw the cue "A" followed by probe "X," they were to respond by pressing the "yes" button on the keyboard. For any other possible cue-probe combination, participants were told to press the "no" button on the keyboard. Demands for goal maintenance were introduced by presenting three distractor letters between every cue-probe pair. The cues and the probes were red, whereas the distractors (any letter except "A," "X," and "Y") were always white. Distractors were each presented for 300 ms with a 1,200 ms interval between letters. Participants were to respond with a "no" button press to the distractors. Each session started with the practice block. The practice block was made up of 10 trials including all four possible experimental conditions: AX (an "A" cue followed by an

"X" probe); BX (an "X" probe preceded by a non-A-cue); AY (any probe but "X" preceded by the letter "A"); and BY (any cue but "A" and any probe but "X"). Participants were provided with feedback on accuracy and RT after each practice trial. In both practice and experimental phases, AX trials occurred for 70% of the time, whereas each of the remaining experimental conditions appeared for 10% of the time. Previous studies have employed high-conflict trials (BX, AY) to examine differences in proactive and reactive control structures [15,26,27]. Baseline measurements in AX-CPT include mean RT and accuracy computed for each trial [28]. Chiew and Braver [29] introduced performance in the AY and BX trials as the proactive control Index and reactive control Index, respectively. These indices are derived from mean error and RT for correct responses [15].

## Procedure

The study took place in a controlled laboratory setting, with participants seated in the experimental room at Tehran University. Upon receiving their consent, they were briefed on the tasks. Initially, they were introduced to the Go/No-Go task, designed to assess single-mode cognitive control [25] (Fig 1A). Following that, they were provided with an explanation of the AX-CPT task, which measures dual mechanisms of cognitive control [26,27] (Fig 2A). This study commenced on 07/01/2019 and concluded on 11/30/2019.

## Data analysis

The statistical analysis plan encompassed a comprehensive approach to investigate the effects of various SOAs and trial types on RT and accuracy across different tasks. Initially, we employed separate one-way ANOVA tests to examine how different SOAs influenced RT and accuracy in the Go/No-Go task. This allowed us to discern any significant differences in performance metrics based on the timing of stimulus presentation. Subsequently, a series of one-way ANOVA tests were conducted to explore the impact of different trial types on RT and accuracy within the AX-CPT task. This analysis aimed to identify how variations in task demands and cognitive load affected participants' performance metrics. Furthermore, to provide deeper insights into task-specific differences, additional comparisons of RT and accuracy were carried out using t-tests within each task. These comparisons enabled us to pinpoint specific conditions or trial types that elicited significant differences in behavioral outcomes. Throughout the analyses, all statistical tests were conducted with a predetermined significance level of $P < 0.05$, ensuring robustness and reliability in the interpretation of results. Each step in the statistical analysis plan was designed to elucidate the nuanced effects of SOAs and trial types on cognitive performance, contributing to a comprehensive understanding of task-related dynamics.

## Results

The Go/No-Go task assesses inhibition, where participants must refrain from responding to specific stimuli (No-Go trials) while responding to others (Go trials, Fig 1A). On the other hand, the AX-CPT task evaluates proactive/reactive control, focusing on the maintenance and manipulation of task-relevant information to anticipate and appropriately respond to specific cue-target sequences (Fig 2A). In this study, we employed these tasks to probe cognitive control mechanisms under varying conditions, particularly focusing on the impact of extended SOA in the Go/No-Go paradigm and the emergence of proactive control in the AX-CPT paradigm.

### Enhanced RT with extended stimulus onset asynchrony within single-mode cognitive control

Our findings revealed a notable trend: as the SOAs increased within the context of the Go/No-Go task, a corresponding decrease in RT was observed specifically during the GO trials ($F_{4, 4370}$ = 13.65, p < 0.001, Fig 1B). This suggests a potentially beneficial effect of extended SOAs on task performance, indicating that participants may have had more time to prepare and execute responses during the GO trials, leading to faster RTs. Additionally, prolonged SOA intervals might have facilitated better anticipation and processing of stimulus onset, thereby enhancing overall task efficiency.

A marked contrast in accuracy was evident between Go and No-Go trials across all levels of SOAs, with No-Go trials consistently demonstrating notably higher accuracy rates compared to Go trials ($F_{1, 8740}$ = 145.62, p < 0.001, Fig 1C). This disparity in accuracy highlights the differential cognitive demands associated with responding to Go and No-Go stimuli, with participants exhibiting greater accuracy in inhibiting responses during No-Go trials. Moreover, the statistical analysis revealed no significant dependence on SOA levels ($F_{4, 8740}$ = 0.15, p = 0.96, Fig 1C), suggesting that changes in SOAs did not significantly affect accuracy. This implies that variations in the timing of stimulus presentation did not impact participants' ability to accurately respond to the task stimuli, indicating robust performance across different SOA conditions. Furthermore, the absence of a significant interaction effect between trial types (Go and No-Go) and SOA levels ($F_{4, 8740}$ = 0.40, p = 0.99, Fig 1C) indicates that the relationship between trial types and SOAs remained consistent across all experimental conditions. This consistency suggests that the influence of SOAs on task performance did not differ between Go and No-Go trials, further supporting the stability of task dynamics across varying SOA levels.

### Emergence of proactive control within dual mechanisms of cognitive control

In the AX-CPT task, a thorough investigation uncovered a significant variation in accuracy among different trial types (AX, AY, BX, and BY, $F_{1, 7160}$ = 145.62, p < .001, Fig 2B). Notably, BY trials exhibited markedly higher accuracy rates compared to AX (p = .01), AY (p = .01), and BX (p = .002) trials, indicating a consistent pattern of superior performance in detecting targets. Furthermore, our analysis of RT revealed substantial differences across trial types (AX, AY, BX, and BY, $F_{1, 7160}$ = 145.62, p < .001, Fig 2B). Specifically, the RT in BX trials was notably shorter compared to AX (p = .001), AY (p = .001), and BY (p = 0.02) trials, suggesting heightened efficiency in processing target stimuli. Conversely, the RT in AY trials were notably prolonged compared to BX (p = 0.001), AX (p = 0.001), and BY (p = 0.001) trials. These findings indicate that participants exhibit greater proactive control, as evidenced by poorer AY performance and improved BX performance.

Our results showed that participants demonstrated enhanced proactive control during the AX-CPT task, as indicated by their performance across different trial types. Specifically, participants exhibited poorer performance in AY trials, reflecting their proactive control strategies in maintaining task-relevant information and inhibiting irrelevant responses. Conversely, participants showed improved performance in BX trials, suggesting their ability to efficiently update working memory representations and appropriately respond to target stimuli.

## Discussion

The findings of our study reveal two key observations regarding cognitive control mechanisms in healthy adults: Firstly, our results demonstrate that extending SOA leads to enhanced RT

within single-mode cognitive control in healthy adults (Fig 1). Secondly, our study reveals the emergence of proactive control within the dual mechanisms of cognitive control in healthy adults (Fig 2). Proactive control refers to the anticipation and implementation of cognitive strategies to prepare for upcoming task demands, enabling individuals to proactively regulate their behavior and optimize performance.

Our results demonstrate that extending SOA leads to enhanced RT within single-mode cognitive control in healthy adults, suggesting that underlying neural mechanisms involved in response inhibition and motor preparation may be implicated in this effect. Neuroimaging studies have consistently implicated the prefrontal cortex (PFC), particularly the ventrolateral prefrontal cortex (VLPFC), in the inhibition of prepotent responses during Go/No-Go tasks [30,31]. The VLPFC exerts top-down control over subcortical regions, such as the basal ganglia, which are crucial for motor planning and execution [30,32]. Extended SOAs may provide additional time for preparatory processes mediated by the PFC, thereby facilitating more effective suppression of prepotent responses and faster initiation of motor responses during Go trials [33]. These intervals might enhance the recruitment and coordination of neural networks involved in response selection and execution, thereby improving RTs [34,35]. Moreover, prolonged SOAs may also improve the efficiency of sensory processing and perceptual discrimination, enabling participants to better anticipate and prepare for the onset of Go stimuli (34,35). The observed differential accuracy rates between Go and No-Go trials across all SOA levels highlight the distinct cognitive demands associated with response inhibition and execution. The higher accuracy in No-Go trials suggests successful inhibition of prepotent responses, likely involving activations in the anterior cingulate cortex (ACC) and other regions related to error monitoring and conflict resolution [30,32]. Furthermore, the consistent accuracy rates across different SOA conditions indicate that variations in stimulus timing did not compromise participants' ability to inhibit responses accurately [34,35]. This robust performance across SOA conditions underscores the reliability and stability of inhibitory control processes mediated by the neural pathways underlying response inhibition. From a broader perspective, these findings provide insights into the underlying mechanisms of cognitive control. The enhancement of RT with extended SOAs suggests that temporal factors play a crucial role in the efficiency of cognitive control processes. Specifically, the additional time afforded by longer SOAs appears to support preparatory processes and enhance the coordination of neural networks involved in response selection and execution. Understanding these temporal dynamics not only contributes to theoretical models of cognitive control but also has practical implications. For instance, optimizing SOAs in cognitive tasks could potentially improve performance outcomes in various domains requiring rapid and accurate decision-making, such as sports training, educational settings, and clinical interventions aimed at enhancing cognitive function.

The emergence of proactive control within the dual mechanisms of cognitive control in healthy adults suggests a sophisticated interplay of brain mechanisms underlying adaptive cognitive processes. Our findings indicate that alongside reactive control mechanisms, proactive control strategies are engaged, involving anticipatory adjustments in cognitive processing to optimize performance. Neuroimaging studies have consistently implicated several brain regions in proactive control. Prefrontal regions, such as the dorsolateral prefrontal cortex (DLPFC) and ACC, are known for their roles in goal maintenance, context monitoring, and response selection [2,32]. Activation in these regions likely reflects the engagement of top-down processes to bias attention, inhibit prepotent responses, and maintain task goals, thereby facilitating proactive control [32,36]. Moreover, the emergence of proactive control within the dual mechanisms of cognitive control may entail complex interactions between prefrontal regions and subcortical structures like the basal ganglia and thalamus. The basal ganglia,

crucial for action selection and reinforcement learning, contribute to implementing learned strategies and automatic responses [37]. Proactive control likely relies on integrating prefrontal signals with basal ganglia-mediated action selection mechanisms to bias behavior toward goal-relevant responses [3,37]. Furthermore, the dynamic balance between proactive and reactive control mechanisms may be modulated by neurotransmitter systems, particularly dopamine. Dopaminergic signaling is implicated in regulating cognitive flexibility, reward processing, and modulating prefrontal function [37]. Variability in dopamine levels or receptor availability may influence the propensity to engage in proactive control strategies, with higher dopamine levels associated with increased proactive control and enhanced cognitive flexibility [3,37]. From a practical standpoint, understanding these underlying mechanisms of cognitive control has significant implications. Optimizing the balance between proactive and reactive control strategies could potentially improve performance across various domains, such as education, healthcare, and workplace settings. For example, interventions aimed at enhancing proactive control through cognitive training or pharmacological approaches targeting dopamine systems might benefit individuals requiring sustained attention and efficient decision-making in complex environments.

The study encountered three main limitations. Firstly, the sample size was small. Secondly, conducting brain function assessments using neuroimaging techniques during the Go/No-Go and AX-CPT tasks was not feasible. Future research endeavors may offer valuable insights by addressing these constraints. Lastly, as the sample size was comprised solely of healthy participants, further investigation is warranted in individuals with psychiatric disorders like schizophrenia and those with paranormal beliefs, as they may present cognitive deficits [6,7,38–47] in addition to using neuroimaging techniques.

Future studies should include more diverse samples, including individuals with psychiatric disorders, such as schizophrenia, and those with different cognitive profiles to determine if the findings generalize across different populations. Implementing neuroimaging techniques during tasks like Go/No-Go and AX-CPT could provide direct evidence of the neural mechanisms involved in both single-mode and dual cognitive control processes. Designing and testing cognitive training programs aimed at enhancing proactive control strategies and assessing their efficacy in improving cognitive function in various populations, including aging adults and individuals with cognitive impairments.

In conclusion, our study highlights two significant findings regarding cognitive control mechanisms in healthy adults: firstly, the enhancement of RT with extended SOA within single-mode cognitive control, and secondly, the emergence of proactive control within the dual mechanisms of cognitive control. Overall, these findings highlight the effect of SOA on single-mode cognitive control and the emergence of proactive control within dual mechanisms of cognitive control in healthy adults.

## Supporting information

**S1 Data.**
(XLSX)

## Author Contributions

**Data curation:** Abdolvahed Narmashiri.

**Formal analysis:** Abdolvahed Narmashiri.

**Investigation:** Abdolvahed Narmashiri.

**Methodology:** Abdolvahed Narmashiri.

**Project administration:** Abdolvahed Narmashiri.

**Resources:** Abdolvahed Narmashiri.

**Software:** Abdolvahed Narmashiri.

**Supervision:** Abdolvahed Narmashiri.

**Validation:** Abdolvahed Narmashiri.

**Visualization:** Abdolvahed Narmashiri.

**Writing – original draft:** Abdolvahed Narmashiri.

**Writing – review & editing:** Abdolvahed Narmashiri.

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
