## [Decision Letter · Decision Letter 0]

12 Jun 2024

PONE-D-24-17692

Manipulations of Stimulus Onset Asynchrony Alter Cognitive Control in Healthy Adults

PLOS ONE

Dear Dr. Narmashiri,

Thank you for submitting your manuscript to PLOS ONE. After careful consideration, we feel that it has merit but does not fully meet PLOS ONE’s publication criteria as it currently stands. Therefore, we invite you to submit a revised version of the manuscript that addresses the points raised during the review process.

We look forward to receiving your revised manuscript.

Kind regards,

Rasool Abedanzadeh, Ph.D

Academic Editor

PLOS ONE

Journal Requirements:

Additional Editor Comments:

Dear Author,

Please revise your manuscript according to the reviewers' comments.

Reviewers' comments:

Reviewer's Responses to Questions

**Comments to the Author**

1. Is the manuscript technically sound, and do the data support the conclusions?

Reviewer #1: Yes

Reviewer #2: Yes

Reviewer #3: Yes

2. Has the statistical analysis been performed appropriately and rigorously? 

Reviewer #1: Yes

Reviewer #2: Yes

Reviewer #3: Yes

3. Have the authors made all data underlying the findings in their manuscript fully available?

Reviewer #1: Yes

Reviewer #2: No

Reviewer #3: Yes

4. Is the manuscript presented in an intelligible fashion and written in standard English?

Reviewer #1: Yes

Reviewer #2: Yes

Reviewer #3: Yes

5. Review Comments to the Author

Reviewer #1: Dear Author,

The manuscript entitled "Manipulations of Stimulus Onset Asynchrony Alter Cognitive Control in Healthy Adults" presents a topic that may interest readers of the journal. Although I consider an interesting topic, I would like to make some comments about the present manuscript below:

Introduction

1. The use of terms like "single-mode cognitive control" and "dual mechanisms of cognitive control" should be briefly defined for clarity, as not all readers might be familiar with these concepts.

Methods

1. While the methods are described, more detail could be provided on the sample size and characteristics, as well as the specific tasks used (Go/No-Go task and AX-CPT task), to give readers a better understanding of the experimental setup.

2. The description of participants is thorough, but it would be helpful to include more information on how participants were recruited and any exclusion criteria beyond the absence of specific conditions.

3. Good inclusion of ethical considerations, mentioning adherence to the Declaration of Helsinki and informed consent. This ensures ethical transparency.

4. The methods section is detailed and comprehensive, which is excellent. The explanation of the Cue-dependent Go/No-Go task is clear and includes necessary details on task design, cue presentation, and target stimuli.

5. The design of the task, including the use of various SOAs and the randomization process, is well-explained. However, the rationale for choosing specific SOA values (100, 200, 300, 400, and 500 ms) could be provided to help readers understand the experimental choices.

6. The description of how reaction times and accuracy were recorded is clear. However, mentioning any software or hardware specifics (e.g., type of keyboard, response time accuracy of the computer) can add to the reproducibility of the study.

7. While the data collection methods are well-detailed, the methods section could be improved by including a brief description of the statistical analysis plan.

Results nd discussion

1. The results are well-presented, but the discussion on their implications could be expanded. For example, what do these findings suggest about the underlying mechanisms of cognitive control and their practical applications?

2. The discussion could mention potential future research directions or practical applications of the findings, which would provide a sense of how this study contributes to the broader field of cognitive control research.

Reviewer #2: This study is investigating the effect of stimulus onset asynchrony and strategy training on single-mode cognitive control (Go/No Go task) and dual mechanism of cognitive control (AX-CPT task).

In this study, no clinical trials were conducted, and no interventions were administered. Instead, participants completed two computerized cognitive tasks after providing written informed consent before enrollment.

In my opinion, the topic of this research is interesting but lacks innovation. Since the effect of stimulus onset asynchrony on cognitive control has been investigated in past researches, the main goal of the current research is not clear and the introduction of the research has not been able to determine the turning point of the work. Although this research is a basic research, the applications of the results have not been determined. In addition, the research methodology also lacks innovation.

Unfortunately, due to lake of innovation in the subject and method, I suggest the non-acceptance of the work for publication.

Reviewer #3: A. Hashemi, [۰۹.۰۶.۲۴ ۰۰:۱۴]

1. The title of the research is presented in a biased way, so it should be corrected.

2. In the abstract section, the exact P value should be mentioned.

3. The gap and necessity of research in relation to this issue should be expressed in a more methodical way

4. The method section is presented after the findings. Is there a specific reason?

5. A lot of time has passed from the time of the research (2019) to the presentation of the article (2024). Is there a specific reason?

6. The location of the research should be mentioned.

7. The discussion section is very long. be presented in a more concise and useful manner.

8. In the conclusion section, mention the general conclusion of the research in one to several lines.

6. PLOS authors have the option to publish the peer review history of their article (what does this mean?). If published, this will include your full peer review and any attached files.

Reviewer #1: **Yes: **Ebrahim Norouzi

Reviewer #2: No

Reviewer #3: No

---

## [Author Response · Author response to Decision Letter 0]

15 Jun 2024

Response to reviewers

Reviewers' comments:

Reviewer's Responses to Questions

Comments to the Author

1. Is the manuscript technically sound, and do the data support the conclusions?

Reviewer #1: Yes

Reviewer #2: Yes

Reviewer #3: Yes

Response: Thank you for reviewers and editor comment. We have meticulously addressed all comments from the editor and reviewers. 

2. Has the statistical analysis been performed appropriately and rigorously?

Reviewer #1: Yes

Reviewer #2: Yes

Reviewer #3: Yes

Response: Thank you

3. Have the authors made all data underlying the findings in their manuscript fully available?

Reviewer #1: Yes

Reviewer #2: No

Reviewer #3: Yes

Response: I have added the Data Availability section to the manuscript on page 22, lines 5.

“[Data Availability: Data available on request from the author.]”

4. Is the manuscript presented in an intelligible fashion and written in standard English?

Reviewer #1: Yes

Reviewer #2: Yes

Reviewer #3: Yes

Response: Thank you. 

5. Review Comments to the Author

Response to Reviewer #1

Dear Author,

The manuscript entitled "Manipulations of Stimulus Onset Asynchrony Alter Cognitive Control in Healthy Adults" presents a topic that may interest readers of the journal. Although I consider an interesting topic, I would like to make some comments about the present manuscript below:

Response: Thank you for your insightful feedback. We appreciate your suggestions and hope that our revised manuscript comprehensively addresses these concerns throughout the manuscript. Accordingly, we have revised the entire manuscript, including the abstract, introduction, methods, and discussion, to address these concerns. 

Introduction

1. The use of terms like "single-mode cognitive control" and "dual mechanisms of cognitive control" should be briefly defined for clarity, as not all readers might be familiar with these concepts.

Response: Based on this valuable feedback, I have revised the Introduction section on page 2, lines 1-15.

“[Cognitive control, comprising both single-mode and dual mechanisms, serves as a fundamental aspect of executive functioning, allowing individuals to regulate behavior and decision-making processes 1-3. Single-mode cognitive control refers to a uniform strategy used to manage cognitive processes, often assessed through tasks like the Go/No-Go paradigm, which involves the ability to inhibit automatic responses and maintain task goals amidst distracting stimuli 4-7. Conversely, dual mechanisms of cognitive control encompass two distinct modes: proactive control, which involves the anticipatory allocation of cognitive resources, and reactive control, which involves responding to immediate demands. These dual mechanisms are examined in tasks like the AX-Continuous Performance Task (AX-CPT), which requires the interplay of proactive and reactive strategies to optimize performance in challenging cognitive contexts 3,8. The efficiency of cognitive control in healthy adults can be influenced by various factors, including the stimulus onset asynchrony (SOA) effect and strategy training.]”

Methods

1. While the methods are described, more detail could be provided on the sample size and characteristics, as well as the specific tasks used (Go/No-Go task and AX-CPT task), to give readers a better understanding of the experimental setup.

Response: I have revised the Materials and Methods section based on your valuable comment, as detailed on pages 5-11.

“[Materials and Methods 

Participants

Thirty-four healthy students (19 females and 15 males), who were determined to be right-handed using the Edinburgh Handedness Inventory, were recruited for this research. Participants were selected based on their availability and willingness to participate. Exclusion criteria included self-reported history of psychosis, mental disorders, acute or chronic illnesses, neurological or personality issues, substance abuse, or epilepsy. Participants also reported normal or corrected-to-normal vision. Power analysis using G*Power 3.1 [22] with a medium effect size, power of 0.85, and α = 0.05, suggested a sample size of thirty-one, with slight oversampling to mitigate potential technical issues. Recruitment took place through advertisements on campus and social media platforms. Participants were compensated with gifts for their involvement. In this study, supported by the Institute for Cognitive Science Studies, no clinical trials were conducted, and no interventions were administered. Instead, participants completed two computerized cognitive tasks after providing written informed consent before enrollment. The study adhered to the principles outlined in the Declaration of Helsinki.

Stimuli. 

Cue-dependent Go/No-Go task

This task assesses cognitive control in a specific mode, emphasizing the predictive elements of both inhibitory and activational control systems. Prior cues inform participants about the upcoming type of stimulus (go or stop) with high accuracy. Participants' inhibitory and activational tendencies adjust rapidly based on these cues, initiating preparatory actions for either restraining or executing a response [23, 24]. The conditions with go cues are especially intriguing as they prompt a strong readiness to respond, resulting in quicker reactions to go targets. Yet, participants need to counteract this readiness to suppress their response when presented with a no-go target. Interestingly, there are more instances of failure to inhibit responses to no-go targets after go cues compared to after no-go cues, indicating a heightened challenge in inhibiting pre-existing responses [24]. Moreover, the ability to exert inhibitory control in the presence of prepotent go cues seems significantly affected by psychoactive drugs, encompassing both stimulants and depressants [25]. The study employed Inquisit software (version 6.6.1) running on a personal computer. Each trial followed a specific sequence: (a) a fixation point (+) shown for 800 ms; (b) a blank white screen displayed for 500 ms; (c) presentation of a cue with one of five stimulus onset asynchronies (SOAs = 100, 200, 300, 400, and 500 ms); (d) appearance of a go or no-go target, remaining visible until a response occurred or 1000 ms had passed; and (e) a 700 ms intertrial interval.

The cue, which measured 7.5 × 2.5 cm, was shown as a black-outlined rectangle (0.8 mm thick) centrally displayed on a white background of a computer monitor. This cue could appear either horizontally (2.5 × 7.5 cm) or vertically (7.5 × 2.5 cm). Go targets, colored green, and no-go targets, colored blue, were displayed as solid colors within the rectangle cue. Participants were instructed to press the forward slash (/) key on the keyboard upon seeing a green go target and to refrain from responding when a blue no-go target appeared. Participants used their preferred hand's index finger to respond. Go and no-go targets were presented in different colors, and the cue's orientation indicated the probability of encountering either type of target. Vertical cues suggested an 80% chance of encountering a go target and a 20% chance of a no-go target, while horizontal cues indicated an 80% probability of encountering a no-go target and a 20% probability of encountering a go target. Therefore, vertical cues indicated a need for a go response, while horizontal cues signaled a requirement for a no-go response. Changing SOAs between cues and targets encouraged participants to focus on the cues, and the random nature of SOAs prevented anticipation of when the targets would appear. The experiment consisted of 250 trials covering all possible combinations of cues and targets. There were an equal number of vertical and horizontal cues (125 each), each preceding an equal number of go and no-go target stimuli (125 each). Every cue-target combination was presented at each of the five SOAs, with an equal distribution of SOAs between pairs. The order of presentation for cue-target pairs and SOAs was randomized. The computer recorded responses during each trial, measuring reaction times (RT) from target onset to key press in ms. On average, completing the test took approximately 10 minutes (Fig. 1A).

The task design incorporates various SOA values (100, 200, 300, 400, and 500 ms) strategically chosen to probe different aspects of cognitive processing [23, 24]. Shorter SOAs (100 and 200 ms) typically assess automatic or pre-attentive processing, where stimuli are closely spaced, allowing minimal time for deliberate attentional shifts. Intermediate SOAs (300 and 400 ms) explore more controlled processing, balancing between automatic and effortful attentional mechanisms. Longer SOAs (500 ms) emphasize sustained attention and decision-making processes, reflecting scenarios where stimuli are presented further apart, requiring sustained attention to maintain task performance. This range of SOAs enables the examination of how temporal factors influence cognitive processing speed, accuracy, and the efficiency of executive functions such as attentional control and response inhibition.

AX-continuous performance task (AX-CPT).

AX-CPT is utilized as a behavioral measure to assess the dual mechanisms of control (DMC) [26, 27]. For this procedure, participants were presented with a sequence of letters for 300 ms each in the center of a black screen. The letters were displayed on a cue-probe basis so that 5,700 ms elapsed between presentation of cue and probe. The intertrial interval was 1,200 ms. Participants were instructed to maintain the cue in memory (either the letter “A” or any other letter except “X,” “K,” and “Y,” due to their perceptual similarity with “X”) until they saw the probe (either the letter “X” or any other letter except “A,” “K,” or “Y”). Whenever they saw the cue “A” followed by probe “X,” they were to respond by pressing the “yes” button on the keyboard. For any other possible cue-probe combination, participants were told to press the “no” button on the keyboard. Demands for goal maintenance were introduced by presenting three distractor letters between every cue-probe pair. The cues and the probes were red, whereas the distractors (any letter except “A,” “X,” and “Y”) were always white. Distractors were each presented for 300 ms with a 1,200 ms interval between letters. Participants were to respond with a “no” button press to the distractors. Each session started with the practice block. The practice block was made up of 10 trials including all four possible experimental conditions: AX (an “A” cue followed by an “X” probe); BX (an “X” probe preceded by a non-A-cue); AY (any probe but “X” preceded by the letter “A”); and BY (any cue but “A” and any probe but “X”). Participants were provided with feedback on accuracy and RT after each practice trial. In both practice and experimental phases, AX trials occurred for 70% of the time, whereas each of the remaining experimental conditions appeared for 10% of the time. Previous studies have employed high-conflict trials (BX, AY) to examine differences in proactive and reactive control structures [15, 26, 27]. Baseline measurements in AX-CPT include mean RT and accuracy computed for each trial [28]. Chiew and Braver (29) introduced performance in the AY and BX trials as the proactive control Index and reactive control Index, respectively. These indices are derived from mean error and RT for correct responses [15].

Procedure

The study took place in a controlled laboratory setting, with participants seated in the experimental room at Tehran University. Upon receiving their consent, they were briefed on the tasks. Initially, they were introduced to the Go/No-Go task, designed to assess single-mode cognitive control [25] (Fig 1A). Following that, they were provided with an explanation of the AX-CPT task, which measures dual mechanisms of cognitive control [26, 27] (Fig 2A). This study commenced on 07/01/2019 and concluded on 11/30/2019.

Data Analysis

The statistical analysis plan encompassed a comprehensive approach to investigate the effects of various SOAs and trial types on RT and accuracy across different tasks. Initially, we employed separate one-way ANOVA tests to examine how different SOAs influenced RT and accuracy in the Go/No-Go task. This allowed us to discern any significant differences in performance metrics based on the timing of stimulus presentation. Subsequently, a series of one-way ANOVA tests were conducted to explore the impact of different trial types on RT and accuracy within the AX-CPT task. This analysis aimed to identify how variations in task demands and cognitive load affected participants' performance metrics. Furthermore, to provide deeper insights into task-specific differences, additional comparisons of RT and accuracy were carried out using t-tests within each task. These comparisons enabled us to pinpoint specific conditions or trial types that elicited significant differences in behavioral outcomes. Throughout the analyses, all statistical tests were conducted with a predetermined significance level of P < 0.05, ensuring robustness and reliability in the interpretation of results. Each step in the statistical analysis plan was designed to elucidate the nuanced effects of SOAs and trial types on cognitive performance, contributing to a comprehensive understanding of task-related dynamics.]”

2. The description of participants is thorough, but it would be helpful to include more information on how participants were recruited and any exclusion criteria beyond the absence of specific conditions.

Response: Thank. 

3. Good inclusion of ethical considerations, mentioning adherence to the Declaration of Helsinki and informed consent. This ensures ethical transparency.

Response: Thank you. 

4. The methods section is detailed and comprehensive, which is excellent. The explanation of the Cue-dependent Go/No-Go task is clear and includes necessary details on task design, cue presentation, and target stimuli.

Response: Thank you. 

5. The design of the task, including the use of various SOAs and the randomization process, is well-explained. However, the rationale for choosing specific SOA values (100, 200, 300, 400, and 500 ms) could be provided to help readers understand the experimental choices.

Response: I have added new information to address this valuable comment on page 8, lines 3-14.

“[The task design incorporates various SOA values (10

---

## [Decision Letter · Decision Letter 1]

21 Jun 2024

Effects of Stimulus Onset Asynchrony on Cognitive Control in Healthy Adults

PONE-D-24-17692R1

Dear Dr. Narmashiri,

We’re pleased to inform you that your manuscript has been judged scientifically suitable for publication and will be formally accepted for publication once it meets all outstanding technical requirements.

Kind regards,

Rasool Abedanzadeh, Ph.D

Academic Editor

PLOS ONE

Additional Editor Comments (optional):

Dear Author,

Thanks for your corrections and revising the manuscript.

All the reviewers' concerns are addressed.

Reviewers' comments:

Reviewer's Responses to Questions

**Comments to the Author**

1. If the authors have adequately addressed your comments raised in a previous round of review and you feel that this manuscript is now acceptable for publication, you may indicate that here to bypass the “Comments to the Author” section, enter your conflict of interest statement in the “Confidential to Editor” section, and submit your "Accept" recommendation.

Reviewer #1: All comments have been addressed

Reviewer #3: All comments have been addressed

2. Is the manuscript technically sound, and do the data support the conclusions?

Reviewer #1: Yes

Reviewer #3: Partly

3. Has the statistical analysis been performed appropriately and rigorously? 

Reviewer #1: Yes

Reviewer #3: Yes

4. Have the authors made all data underlying the findings in their manuscript fully available?

Reviewer #1: Yes

Reviewer #3: Yes

5. Is the manuscript presented in an intelligible fashion and written in standard English?

Reviewer #1: Yes

Reviewer #3: Yes

6. Review Comments to the Author

Reviewer #1: Dear Authors,

I hope this message finds you well. It was a pleasure to review the revised manuscript. I really like this study. This is a well planned and good written study. Now, Author after revision, have addressed my comments, in my opinion, the manuscript would be ok for publication.

Best regards,

Reviewer #3: Hello. The mentioned article is a valuable research that has met the requirements for publication to a large extent and has answered the opinions of the judges correctly.

7. PLOS authors have the option to publish the peer review history of their article (what does this mean?). If published, this will include your full peer review and any attached files.

Reviewer #1: **Yes: **Ebrahim Norouzi

Reviewer #3: No

---

## [Editor Report · Acceptance letter]

7 Jul 2024

PONE-D-24-17692R1 

PLOS ONE

Dear Dr. Narmashiri, 

I'm pleased to inform you that your manuscript has been deemed suitable for publication in PLOS ONE. Congratulations! Your manuscript is now being handed over to our production team.

Kind regards, 

on behalf of

Dr. Rasool Abedanzadeh 

Academic Editor

PLOS ONE